# Association between dietary inflammatory index and NT-proBNP levels in US adults: A cross-sectional analysis

**Teng-Chi Ma** [1], **Feng Gao** [2], **Xin-Lu Liu** [2], **Chen-Xi Wang** [2], **Qiang Liu** [2], **Jing Zhou** [2]*

**1** The First Affiliated Hospital of Xi'an Jiaotong University, Yulin Hospital, Yulin, Shaanxi, China,
**2** Department of Cardiology, The Affiliated Hospital of Yan'an University, Yan'an, Shaanxi, China

* 25891082@qq.com

## Abstract

### Background

With cardiovascular diseases standing as a leading cause of mortality worldwide, the interplay between diet-induced inflammation, as quantified by the Dietary Inflammatory Index (DII), and heart failure biomarker NT-proBNP has not been investigated in the general population.

### Methods

This study analyzed data from the National Health and Nutrition Examination Survey (NHANES) 1999–2004, encompassing 10,766 individuals. The relationship between the DII and NT-proBNP levels was evaluated through multivariable-adjusted regression models. To pinpoint crucial dietary components influencing NT-proBNP levels, the LASSO regression model was utilized. Stratified analyses were then conducted to examine the associations within specific subgroups to identify differential effects of the DII on NT-proBNP levels across diverse populations.

### Results

In individuals without heart failure, a unit increase in the DII was significantly associated with an increase in NT-proBNP levels. Specifically, NT-proBNP levels rose by 9.69 pg/mL (95% CI: 6.47, 12.91; p < 0.001) without adjustments, 8.57 pg/mL (95% CI: 4.97, 12.17; p < 0.001) after adjusting for demographic factors, and 5.54 pg/mL (95% CI: 1.75, 9.32; p = 0.001) with further adjustments for health variables. In participants with a history of heart failure, those in the second and third DII quartile showed a trend towards higher NT-proBNP levels compared to those in the lowest quartile, with increases of 717.06 pg/mL (95% CI: 76.49–1357.63, p = 0.030) and 855.49 pg/mL (95% CI: 156.57–1554.41, p = 0.018). Significant interactions were observed in subgroup analyses by age (<50: β = 3.63, p = 0.141; 50–75: β = 18.4, p<0.001; >75: β = 56.09, p<0.001), gender (men: β = 17.82, p<0.001; women: β = 7.43, p = 0.061), hypertension (β = 25.73, p<0.001) and diabetes (β = 38.94, p<0.001).

**Data Availability Statement:** All data used in this study are publicly available in the NHANES database, https://wwwn.cdc.gov/nchs/nhanes/Default.aspx.

**Funding:** The study was funded by the Yan'an Science and Technology Plan Project (Grant No. 2022SLSFGG-025).The funders had no role in study design, data collection and analysis, decision to publish, or preparation of the manuscript.The authors declare no conflicts of interest and confirm the originality of the work.

**Competing interests:** The authors have declared that no competing interests exist.

## Conclusion

This study identified a positive correlation between the DII and NT-proBNP levels, suggesting a robust link between pro-inflammatory diets and increased heart failure biomarkers, with implications for dietary modifications in cardiovascular risk management.

## 1. Introduction

Over recent decades, cardiovascular diseases (CVD) have risen to become a leading global health concern. The World Health Organization identifies CVD as the primary cause of mortality worldwide, responsible for millions of fatalities each year [1,2]. The surge in heart failure (HF) cases, propelled by an aging demographic, highlights the critical need for extensive research into cardiovascular health. However, HF frequently develops insidiously, with significant symptoms emerging only upon the occurrence of severe health episodes. Over time, the heart experiences alterations due to myocardial cell remodeling. In older individuals, this remodeling can result in a progressive deterioration of heart function, ultimately leading to HF [3]. Therefore, identifying risk factors for HF is essential for its prevention, early diagnosis, and management.

Chronic inflammation is recognized as a pivotal risk factor in the intricate causality of CVDs [4,5]. Recent investigations into the Dietary Inflammatory Index (DII) have revealed a significant association between the DII and cardiovascular health [6,7]. The DII evaluates the inflammatory impact of dietary components, offering insights into chronic inflammation and the risk of CVD [8]. Since its inception in 2009 and subsequent update in 2014, the DII has individually tested the diet-induced inflammatory potential [9,10]. The DII objective is to provide a comprehensive analysis of the inflammatory effects of various dietary components, thereby determining the overall inflammatory potential of the diet.

Conventionally, the diagnosis of HF has relied on symptoms and sign evaluation. However, the last two decades have seen N-terminal pro B-type natriuretic peptide (NT-proBNP) measurement emerge as a superior diagnostic criterion for HF [11,12]. NT-proBNP, a sensitive indicator of cardiac strain and dysfunction, is closely associated with the incidence and severity of HF. Its role extends to monitoring therapeutic outcomes and predicting the disease prognosis [13–16]. offering a dynamic and precise measure of cardiac health status beyond conventional diagnostic practices.

This study aimed to investigate the association between the DII and NT-proBNP levels utilizing data from the National Health and Nutrition Examination Survey (NHANES, 1999–2004).

## 2 Methods

The NHANES protocol and stored serum research program were approved by the National Center for Health Statistics Ethics Review Board, and written informed consent was obtained from all participants. All data were anonymized during the collection and processing stages, ensuring that
neither the authors nor the research team had access to any information that could potentially identify
individual participants.

## Study design

This study sourced data from the NHANES, a comprehensive database maintained by the Centers for Disease Control and Prevention (CDC) [17] that encompasses health and medical records of over 120,000 U.S. civilians from 1998 to 2022. The dataset contains over 1,000 variables, including demographic information, results from physical examinations, and laboratory tests.

This study integrated data from three NHANES cycles (1999–2000, 2001–2002, and 2003–2004) to evaluate NT-proBNP levels in participants at a mobile examination center, with blood samples preserved for future analysis. Participants were excluded based on the following criteria: out-of-range NT-proBNP levels (n = 365), incomplete dietary data (n = 528), self-reported pregnancy (n = 627), and absence of HF diagnosis data (n = 43), the study population was narrowed down to 10766 individuals, comprising 5475 women and 5313 men. In the women cohort, 2,552 participants were under 50, 2117 were aged between 50 and 75, and 784 were over 75. Among the men, 2426 were younger than 50, 2159 were within the 50 and 75 age range, and 728 were older than 75. The flowchart (Fig 1) was developed to outline the participant selection.

## Assessment of NT-proBNP levels

Serum samples from the 1999–2004 NHANES cycles were analyzed for NT-proBNP levels, with the assays conducted between 2018 and 2020 at the University of Maryland School of Medicine's laboratory. The NHANES Biospecimen Program provided protocols for sample handling, including details on processing and storage [18].NT-proBNP measurements utilized the Roche Cobas e601 analyzer (Elecsys, Roche Diagnostics), adhering to established detection limits (5 pg/mL to 35,000 pg/mL) and demonstrating a coefficient of variation of 3.1% at a low concentration (46 pg/mL) and 2.7% at a high concentration (32,805 pg/mL). A consistent detection threshold was employed to guarantee the accuracy and reliability of the measurements [19]. Furthermore, prior research has demonstrated the stability of NT-proBNP in samples stored over extended periods [20–22].

## Assessment of dietary inflammatory index

The DII was the critical measure for evaluating the inflammatory potential of participants' diets. The DII quantifies dietary inflammatory potential, ranging from anti-inflammatory to pro-inflammatory effects. This metric, thoroughly validated by Shivappa et al [10], was applied to dietary data collected through 24-hour nutritional recalls at the Mobile Examination Center (MEC). Shivappa and colleagues have demonstrated that using a maximum of 30 food parameters is sufficient to maintain the predictive accuracy of the DII for diet-related inflammation [23]. Participants' intake of foods and beverages was meticulously recorded, with the United States Department of Agriculture (USDA) Food and Nutrient Database for Dietary Studies used to quantify nutrient content. The DII calculation in this research was based on 27 dietary components known to modulate inflammation, including energy, protein, carbohydrate, fiber, total fatty acid, total saturated fatty acid, monounsaturated and polyunsaturated fatty acids, cholesterol, a range of vitamins and minerals including vitamin A, β-carotene, vitamin B1, vitamin B2, niacin, vitamin B6, folate, vitamin B12, vitamin C, vitamin E, magnesium, iron, zinc, and selenium, caffeine, alcohol, and n-3 and n-6 fatty acids. Each participant's dietary intake was matched to a comprehensive database, encompassing global average daily intake figures and their standard deviations for each dietary component across 45 populations worldwide. The intake of each nutrient was contrasted with its global average and subsequently divided by the standard deviation to calculate the Z-scores for each nutrient. These Z-scores

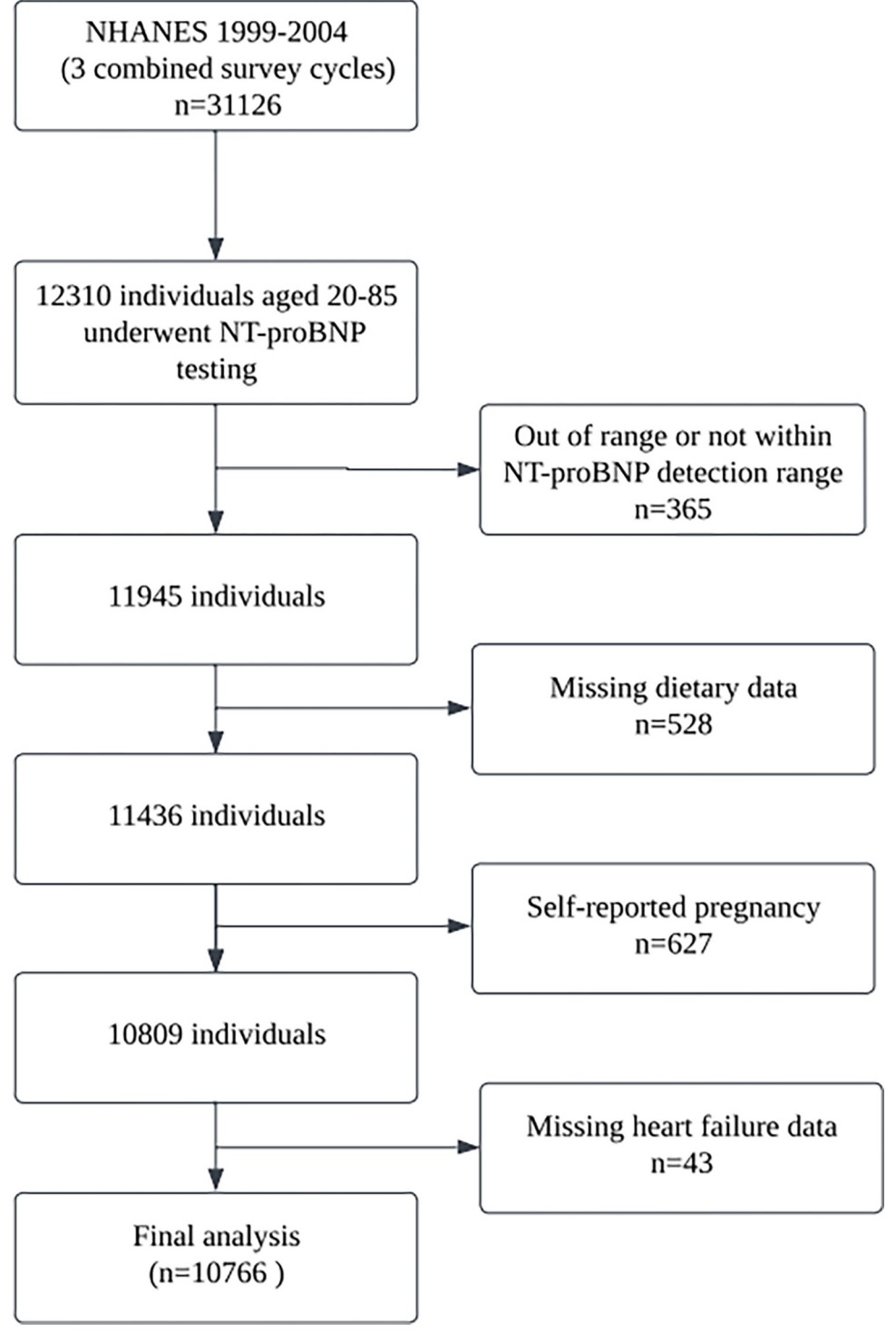

**Fig 1. Flow chart of study participants.**

were converted into percentiles, centered around zero by doubling and then subtracting one, yielding scores from -1 to +1. Each centered score was multiplied by its respective dietary component's inflammation effect score to calculate specific DII scores for each food parameter

[10]. The overall DII score for an individual was calculated by aggregating these specific DII scores. Within this research, the range of the DII scores spanned from a minimum of -4.28 to a maximum of 5.18.

## Assessment of study participants' characteristics

Sociodemographic characteristics captured details including age, gender, ethnic and racial identity (classified as Non-Hispanic White, Mexican American, Non-Hispanic Black, and other) and physical measurements such as waist circumference and BMI. Medical and lifestyle elements brought into the analysis included hypertension—ascertained by a SBP of 140 mmHg or more, or a DBP of 90 mmHg or more, or the administration of antihypertensive drugs. The criterion for diabetes included a self-reported diagnosis or an HbA1c exceeding 6.5%. The diagnosis of hyperlipidemia rested on having total cholesterol readings of 200 mg/dL or higher, or the use of specific medications, or a previously established diagnosis. Renal health was gauged using the updated Chronic Kidney Disease Epidemiology Collaboration formula. Individuals presenting an ACR above 30 mg/g or an eGFR below 60 mL/min/1.73m$^2$ were classified as CKD patients, as per the comprehensive definitions [24]. Cardiovascular conditions were determined from self-disclosed histories of medically confirmed conditions such as congestive heart failure. Smoking behaviors adhered to the NCHS's categorical divisions: never, former, or current smokers. Alcohol consumption delineations encompassed non-drinkers (fewer than 12 lifetime alcoholic beverages), previous drinkers (a year or more of abstinence after previous consumption), light drinkers (a daily intake of up to one drink for women and two for men), moderate drinkers (daily intake of up to three drinks for women and four for men), and heavy drinkers (daily intake of four or more drinks for women and five or more for men) measured against the previous year's consumption patterns.

## Statistical analysis

Statistical analyses adhered to the protocols established by the CDC [25]. The analysis accounted for the study complex survey design by integrating appropriate sample weights, ensuring the statistical integrity and representativeness of the findings. Continuous and categorical variables were examined using the Student's t-test and chi-square test, respectively, with outcomes reported as actual numbers and weighted percentages. DII scores were categorized into quartiles (Q1: DII $\leq$ 0.321; Q2: 0.321 < DII $\leq$ 1.752; Q3: 1.752 < DII $\leq$ 2.816; Q4: DII > 2.816) to explore the association between DII and NT-proBNP levels.To investigate the effect of an HF history, analyses without adjustment and multivariable-adjusted regression models evaluated differences in continuous variables across DII score quartiles, with results presented as beta coefficients and standard errors. For DII and NT-proBNP level correlation, three regression models were established: a crude model without adjustments, Model I adjusted for age, gender, and race/ethnicity, and Model II further adjusted for hypertension, diabetes,chronic kidney disease (CKD). Stratified analyses were conducted based on age groups (<50 years, 50–75 years, >75 years), gender, race/ethnicity (white or non-white), and the presence of comorbid conditions such as HF, CKD, hypertension, diabetes, and BMI categories (<25, $\geq$25). These analyses aimed to investigate potential significant interactions between the DII and NT-proBNP levels. To identify the most effective dietary predictors of NT-proBNP levels and to mitigate multicollinearity, the Least Absolute Shrinkage and Selection Operator (LASSO) regression model was applied. This approach optimized predictive accuracy by reducing the influence of less significant variables. Cross-validation, involving the subdivision of the dataset into ten portions for repetitive training and evaluation, was employed to refine the model, leading to the selection of optimal parameters. A curve was

plotted to depict model performance across a range of lambda values during the cross-validation process. This plot was generated to identify the lambda value that resulted in the most negligible cross-validation error, a measure known as "minimum deviance." Statistical analyses were executed using the R statistical software package (The R Foundation; version 4.2.0). Statistical significance was set at a P value of <0.05.

## 3 Results

### Baseline characteristics of participants

Table 1 presents the baseline characteristics of the study population, stratified by HF status. Of 10,766 participants, 374 reported a history of HF. The mean age of those with HF was approximately 64.77 ± 1.16 years, with a gender distribution of 48.26% women and 51.74% men. Racial and ethnic backgrounds were predominantly non-Hispanic whites (76.87%), with the remainder being Mexican Americans (3.23%), non-Hispanic blacks (10.76%), and other races (9.14%). The prevalence of CKD, hypertension, hyperlipidemia, and diabetes was significantly higher in the HF group compared to the non-HF group (all p < 0.001). Furthermore, smoking status and drinking habits also differed significantly between the two groups (both p < 0.001), with a higher proportion of former smokers and former drinkers in the HF group.

BMI and waist circumference were significantly higher in the HF group compared to the non-HF group (both p < 0.001). Additionally, the mean DII score was significantly higher in the HF group (1.87 ± 0.09) compared to the non-HF group (1.39 ± 0.04) (p < 0.001).NT-proBNP levels were markedly elevated in the HF group (1035.53 ± 112.38 pg/mL) compared to the non-HF group (115.94 ± 4.88 pg/mL, p < 0.001). Moreover, the HF group had significantly lower serum albumin and HDL-C levels, and higher glycohemoglobin, and C-reactive protein levels (all p < 0.001). LDL-C levels were lower in the HF group (p = 0.012), but total cholesterol levels did not differ significantly between the two groups (p = 0.092).

Similarly, to explore the dietary factors that contributed to the difference of DII between two groups, each component score of DII was presented in S1 Table. We found that the HF group had significantly lower DII scores for energy, protein, carbohydrate, total fatty acid, saturated fatty acid, vitamin B12, selenium, iron, and zinc (P<0.001) compared to the non-heart failure group. On the other hand, the HF group had significantly higher DII scores for dietary fiber, polyunsaturated fatty acids, N-6 fatty acids, vitamin A, vitamin B1, vitamin B2, vitamin B6, vitamin E, folate, niacin, magnesium, and alcohol (P<0.05). In addition, S2 Table shows the differences between the two groups separately for each actual dietary component.

### Associations between Dietary Inflammatory Index (DII) and NT-proBNP levels

Spline smoothing curves illustrated a positive correlation between the DII and NT-proBNP levels (Fig 2). Our regression analyses, presented in Table 2, assessed the association between DII and NT-proBNP concentrations. For participants without a history of HF, an incremental unit in DII was associated with a significant elevation in NT-proBNP levels across different models: an increase of 9.69 pg/mL in the crude model (95% confidence interval [CI]: 6.47, 12.91; p < 0.001), 8.57 pg/mL in Model I (95% CI: 4.97, 12.17; p < 0.001), and 5.54 pg/mL in Model II (95% CI: 1.75, 9.32; p = 0.001) with further adjustments. When DII scores were categorized into quartiles, individuals in the highest quartile (DII≥2.816) had significantly higher NT-proBNP levels compared to those in the lowest quartile (DII<0.321) 43.99pg/mL in the crude model (95%CI: 24.26–63.73, p<0.001), 38.22pg/mL in Model I (95%CI: 16.05–60.38, p = 0.001), and 25.18 pg/mL in Model II (95%CI: 3.04–47.32, p = 0.027).

**Table 1. Baseline characteristics of all participants (N = 10766).**

| Variables | Overall (n = 10766) | Non-HF (n = 10392) | HF (n = 374) | P-value |
|---|---|---|---|---|
| Age(years) | 46.32±0.29 | 45.87±0.28 | 64.77±1.16 | < 0.001 |
| Gender(%) | | | | 0.279 |
| Women | 5453(52.50) | 5284(52.60) | 169(48.26) | |
| Men | 5313(47.50) | 5108(47.40) | 205(51.74) | |
| Race/ethnicity (%) | | | | 0.132 |
| Non-Hispanic White | 5699(73.06) | 5471(72.97) | 228(76.87) | |
| Mexican American | 2373 (6.66) | 2309(6.75) | 64(3.23) | |
| Non-Hispanic Black | 1871(10.29) | 1812(10.28) | 59(10.76) | |
| Other race | 823 (9.99) | 800(10.01) | 23 (9.14) | |
| CKD(%) | 2164(14.14) | 1943(13.25) | 221(51.01) | < 0.001 |
| Hypertension(%) | 4671(35.33) | 4380(34.48) | 291(70.48) | < 0.001 |
| Hyperlipidemia(%) | 8072(73.25) | 7747(72.93) | 325(86.71) | < 0.001 |
| Diabetes(%) | | | | < 0.001 |
| Diabetes | 1516 (9.66) | 1373 (9.06) | 143(34.18) | |
| Prediabetes | 418 (3.07) | 398(3.02) | 20(4.99) | |
| No diabetes | 8832(87.27) | 8621(87.91) | 211(60.83) | |
| Smoking status (%) | | | | < 0.001 |
| Never | 5380(49.85) | 5241(50.20) | 139(35.35) | |
| Former | 2998(25.27) | 2833(24.86) | 165(42.12) | |
| Now | 2388(24.88) | 2318(24.94) | 70(22.52) | |
| Drinking status(%) | | | | < 0.001 |
| Never | 1526(12.30) | 1470(12.27) | 56(13.58) | |
| Former | 2188(16.73) | 2028(16.12) | 160(41.90) | |
| Mild | 3355(32.15) | 3251(32.21) | 104(29.54) | |
| Moderate | 1365(15.20) | 1352(15.50) | 13 (3.07) | |
| Heavy | 1888(19.86) | 1861(20.14) | 27 (8.00) | |
| Not recorded | 444 (3.76) | 430(3.76) | 14(3.89) | |
| Body mass index(kg/㎡) | 28.11±0.11 | 28.05±0.11 | 30.44±0.56 | < 0.001 |
| Waist circumference (cm) | 96.30±0.28 | 96.11±0.27 | 104.72±1.03 | < 0.001 |
| DII | 1.40±0.04 | 1.39±0.04 | 1.87±0.09 | < 0.001 |
| NT-proBNP(pg/mL) | 137.64±5.11 | 115.94±4.88 | 1035.53±112.38 | < 0.001 |
| Albumin(g/dL) | 4.34±0.01 | 4.35±0.01 | 4.16±0.02 | < 0.001 |
| HDL-C (mmol/L) | 1.35±0.01 | 1.35±0.01 | 1.24±0.03 | < 0.001 |
| LDL-C (mmol/L) | 3.12±0.02 | 3.13±0.02 | 2.93±0.08 | 0.012 |
| Triglyceride(mg/dL) | 145.74±2.67 | 144.66± 2.73 | 191.03±10.64 | < 0.001 |
| Total-Cholesterol(mg/dL) | 201.93±0.69 | 202.04±0.70 | 197.14±2.83 | 0.092 |
| Glycohemoglobin (%) | 5.47±0.02 | 5.46±0.02 | 5.99±0.07 | < 0.001 |
| C-reactive protein(mg/dL) | 0.43±0.01 | 0.42±0.01 | 0.77±0.06 | < 0.001 |

Heart failure (HF), Chronic kidney disease (CKD),High-Density Lipoprotein Cholesterol (HDL-C), Low-Density Lipoprotein Cholesterol (LDL-C), Weighted to be nationally representative. Weighted percentage may not sum to 100% because of missing data, Data are presented as mean ± SD or n (weighted %).

In individuals with HF, no significant association was found between DII scores and NT-proBNP levels in the crude model (β = 73.46, 95%CI: -72.99–219.90, p = 0.317), Model I (β = 52.8, 95%CI: -79.85–185.46, p = 0.424), or Model II (β = 106.16, 95%CI: -52.50–264.82, p = 0.182). However, notable differences in NT-proBNP levels were observed among

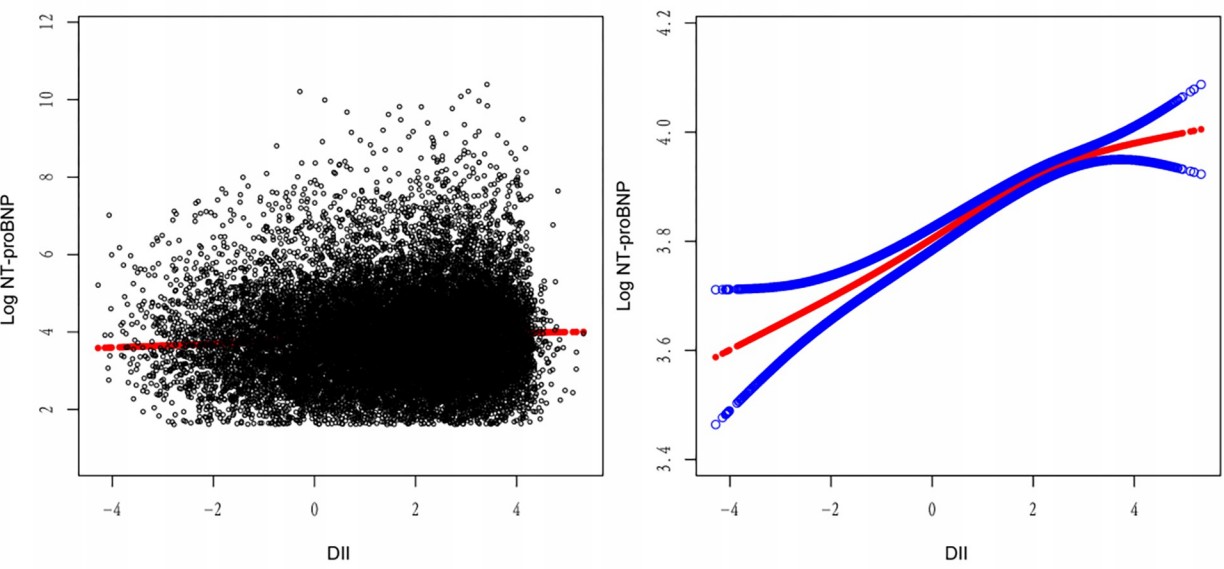

**Fig 2. Dose-response relationship of DII and NT-proBNP levels (log-transformed, pg/dL).** (A) Each dot represents an individual sample. (B) The solid red line indicates the smooth fitting curve between DII and NT-proBNP levels (log-transformed, pg/dL), whereas the dotted blue line represents 95% confidence intervals of the fitting.

participants in the second and third quartiles compared to those in the lowest quartile in Model II, with increases of 717.06 pg/mL (95% CI: 76.49–1357.63, p = 0.030) and 855.49 pg/mL (95% CI: 156.57–1554.41, p = 0.018), respectively.

**Table 2. Relationship between dietary inflammatory index and NT-proBNP.**

| | Crude Model | | Model I | | Model II | |
|---|---|---|---|---|---|---|
| | β (95%CI) | P-value | β (95%CI) | P-value | β (95%CI) | P-value |
| **Non-HF** | | | | | | |
| DII | 9.69(6.47,12.91) | <0.001 | 8.57(4.97,12.17) | <0.001 | 5.54(1.75, 9.32) | 0.001 |
| DII quartile | | | | | | |
| Q1,<0.321 | Reference | | Reference | | Reference | |
| Q2, 0.321–<1.752 | 25.88(-1.88,53.64) | 0.067 | 19.72 (-9.31,48.75) | 0.177 | 16.64(-12.66,45.95) | 0.256 |
| Q3,1.752–<2.816 | 23.12(4.25,41.98) | 0.018 | 15.76 (-5.77,37.29) | 0.146 | 7.85(-14.08,29.78) | 0.471 |
| Q4,≥2.816 | 43.99(24.26,63.73) | <0.001 | 38.22(16.05,60.38) | 0.001 | 25.18(3.04,47.32) | 0.027 |
| P for trend | <0.001 | | 0.002 | | 0.057 | |
| **HF** | | | | | | |
| DII | 73.46(-72.99,219.90) | 0.317 | 52.80 (-79.85,185.46) | 0.424 | 106.16 (-52.50,264.82) | 0.182 |
| DII quartile | | | | | | |
| Q1,<0.321 | Reference | | Reference | | Reference | |
| Q2, 0.321–<1.752 | 485.52(-117.19,1088.23) | 0.111 | 586.65 (-58.94,1232.24) | 0.073 | 717.06(76.49,1357.63) | 0.030 |
| Q3,1.752–<2.816 | 564.09(-99.06,1227.24) | 0.093 | 504.97 (-115.05,1124.99) | 0.107 | 855.49(156.57,1554.41) | 0.018 |
| Q4,≥2.816 | 256.15(-332.46,844.75) | 0.384 | 217.89 (-361.06,796.83) | 0.449 | 453.96 (-215.41,1123.33) | 0.176 |
| P for trend | 0.747 | | 0.972 | | 0.520 | |

Crude Model: No covariates were adjusted.

Model I: Age, gender, race/ethnicity were adjusted.

Model II: Age, gender, race/ethnicity, hypertension, diabetes, chronic kidney disease were adjusted.

## Stratified analysis

This study conducted subgroup analyses (Fig 3) across various predefined strata to identify potential effect modifications between characteristics and outcomes. In our study, age, gender, race, HF, CKD, BMI, hypertension, and diabetes were assessed as potential effect modifiers to determine if the association with the outcome varied significantly across different groups. Across all strata, the association between the DII and NT-proBNP levels was consistently positive (positive β coefficients), and most comparisons within each stratum reached significance (p < 0.05). Every unit increase in DII corresponded to an average rise of 14.32 pg/mL in NT-proBNP levels (95% CI: 9.77, 18.86; p < 0.001). This correlation was robust in older adults, significantly over 75 years (β = 56.09; 95% CI: 29.02, 83.17; p < 0.001), highlighting a significant age-related interaction with the DII and NT-proBNP relationship (p < 0.001). Gender differences were apparent, with a significant association observed in men (β = 17.82; 95% CI: 11.25,

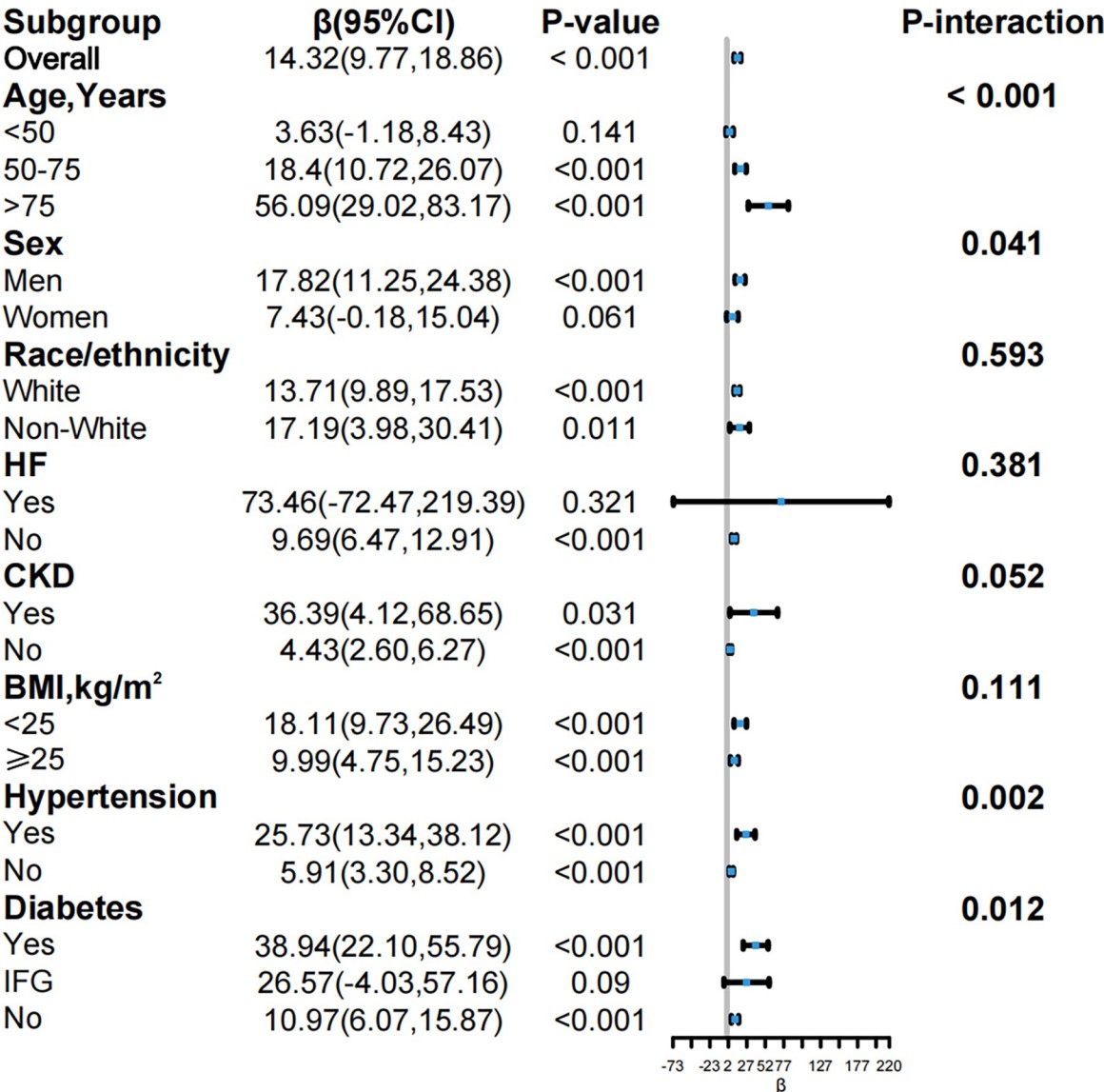

**Fig 3. Forest plots of stratified analyses of DII and NT-proBNP levels.**

24.38; p < 0.001), whereas the association was insignificant in women (β = 7.43; 95% CI: -0.18, 15.04; p = 0.061), and an interaction by gender was observed (p = 0.041). Racial and ethnic analysis indicated positive correlations with NT-proBNP levels for both White and Non-White groups, though no significant interaction was found between these groups (p = 0.593). For individuals with a history of HF, the trend suggested a potential though insignificant association between higher DII and elevated NT-proBNP levels (β = 73.46; 95% CI: -72.47, 219.39; p = 0.321), with no significant interaction observed (p = 0.381). In those with CKD, a significant positive association was observed (β = 36.39; 95% CI: 4.12, 68.65; p = 0.031) alongside a borderline considerable interaction (p = 0.052). The relationship across BMI categories consistently showed a positive trend, though without significant interaction (p = 0.111). A pronounced association was observed in individuals with hypertension (β = 25.73; 95% CI: 13.34, 38.12; p < 0.001), with a significant interaction effect (p = 0.002). Diabetes status further influenced the associations, showing a substantial correlation in those with diabetes (β = 38.94; 95% CI: 22.10, 55.79; p < 0.001) and a significant interaction by diabetes status (p = 0.012).

## Identification of critical dietary factors related to NT-proBNP

A LASSO penalized regression was constructed to identify dietary components intimately associated with NT-proBNP, incorporating 27 dietary factors and three covariates: gender, age, and race or ethnicity (Fig 4). The LASSO technique enhances the robustness of the model by introducing an L1 penalty to the regression, effectively reducing specific coefficients to zero. This approach facilitates feature selection from highly correlated variables, improving the model's interpretability and predictive performance. By eliminating fewer contributory variables, the model enhances its generalization capacity and mitigates the risk of overfitting. The optimization of the loss function, combined with the L1 penalty, leads to a model that

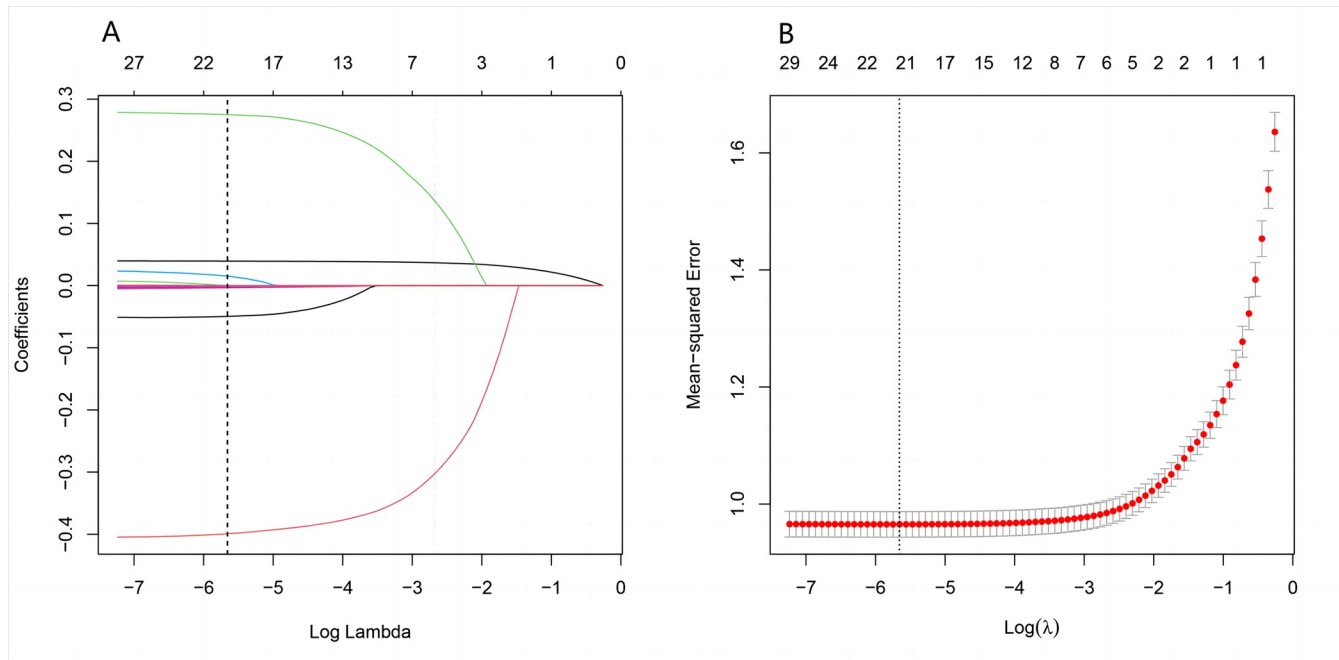

**Fig 4. The LASSO penalized regression analysis for identifying key NT-proBNP -related dietary factors.** (A) The coefficient shrinkage process of all 27 dietary components and 3 covariates (sex, age, and race/ethnicity), we represent the changes in coefficients of different features under various levels of shrinkage by drawing lines of different colors. (B) A 10-fold cross-validation of the LASSO regression model. LASSO, least absolute shrinkage and selection operator.

selectively excludes non-essential predictors, as depicted in Fig 4A.Out of n = 27 dietary variables constituting the DII scores, n = 17 show an interaction to NT-proBNP levels in whole population, such as: Protein, carbohydrates, fiber, monounsaturated and polyunsaturated fatty acids, vitamin A, β-carotene, vitamin B1, niacin, folate, vitamin E, magnesium, iron, selenium, caffeine, alcohol, and n-3 fatty acids, which were identified as the dietary factors substantially correlated with NT-proBNP levels (log-transformed, pg/dL).

## 4 Discussion

This comprehensive, nationally representative cross-sectional study explored the association between the DII and NT-proBNP levels, employing a robust analytical methodology and leveraging a significant sample size. Through applying LASSO penalized regression, dietary factors significantly associated with NT-proBNP levels were identified. Our analysis consistently revealed a positive association between DII and NT-proBNP across various demographic and clinical groups, with a particularly pronounced effect in older adults. This finding highlights the potential of dietary choices on biomarkers of HF, suggesting that nutritional adjustments could play a crucial role in preserving cardiovascular health. Specifically, our results yielded an average increase of 14.32 pg/dL in NT-proBNP for each unit increment in DII (95% CI: 9.77, 18.86; p < 0.001), providing substantive insights into the dietary impact on cardiovascular health, particularly from an inflammatory standpoint.

Beyond focusing on individual foods or nutrients, the complexity of daily diets encompasses combinations of multiple foods and nutrients. In recent years, the significance of diet quality scores and overall dietary patterns, including Healthy Eating Index (HEI), DII, Mediterranean diet (MedDiet), Dietary Approaches to Stop Hypertension (DASH) diet, and vegetarian diets, have been increasingly recognized [26–29]. While previous research on dietary patterns in the context of HF prevention and treatment has primarily focused on the MedDiet and DASH diet [30,31], the role of chronic inflammation in CVD progression has been emphasized [32–34]. Within this framework, dietary patterns emerge as modifiable elements capable of inducing pro- or anti-inflammatory effects. The rich NHANES dataset offers a valuable resource for further investigating this intriguing issue [35].

Our findings are consistent with Liu et al.'s study [36], identifying a positive correlation between the DII and HF(odds ratio [OR] = 1.110, 95% CI: 1.060, 1.163). However, our study extends to investigating the association of DII levels with the cardiovascular health biomarker NT-proBNP rather than solely focusing on HF diagnoses. This approach allows for a more nuanced investigation into the association of diet with cardiovascular health at the biomarker level. Previous studies have highlighted the prevalence of elevated NT-proBNP levels among U.S. adults, even in the absence of a CVD history [37]. Moreover, elevated NT-proBNP levels have been significantly associated with all-cause and CVD mortality among individuals with CVD [38]. Our analysis of the cohort provides deeper insights into the potential mechanisms underlying HF.

Meanwhile, our findings corroborate the case-control study by Jalal Moludi et al. [39], which found that higher DII scores were associated with higher NT-proBNP levels. However, our study benefits from a considerably larger sample size, encompassing 10,766 participants, significantly surpassing Jalal Moludi's study (n = 229). Furthermore, our study's demographic includes a nationally representative sample of Americans, contrasting the Asian population focus in Jalal Moludi's research. This diversity underscores the global relevance of the DII-cardiovascular health connection across different populations, considering variations in regional, racial, lifestyle, and other factors.

Furthermore, subgroup analyses revealed significant interactions between DII and NT-proBNP levels concerning age, gender, hypertension, and diabetes (P for interaction <0.05). Age emerges as a critical factor; with advancing age, the immune system experiences changes that may enhance inflammatory responses [40–42]. Chronic inflammation, mediated by diet and imbalance between pro- and anti-inflammatory factors, could contribute to endothelial injury, vascular remodeling, atherosclerosis, and insulin resistance. These conditions may precipitate alterations in the cardiovascular system, influencing NT-proBNP levels.

The role of gender in this dynamic is significant. Physiological differences, hormone levels, and lifestyle variations between men and women can affect inflammatory responses and cardiovascular function [43,44]. For example, estrogen influences inflammation, and the distinct impacts on cardiovascular metabolism and function suggest that dietary, inflammation, and cardiovascular health interactions manifest differently across sexes [45]. This finding indicates a need for further investigation into the mechanisms behind these sex-specific differences, which would inform targeted nutritional and lifestyle recommendations to enhance cardiovascular health in both sexes.

Chronic conditions such as CKD, hypertension, and diabetes play distinct roles in influencing inflammation status and cardiovascular functionality, potentially impacting NT-proBNP levels. CKD may result in an accumulation of inflammatory markers due to the kidneys' diminished capacity to eliminate metabolites and inflammatory substances [46]. Furthermore, hypertension and diabetes are linked to endothelial dysfunction, which fosters inflammation [47,48]. Moreover, hypertension could trigger cardiac hypertrophy and changes in hemodynamics, influencing NT-proBNP production [49]. Despite variations in absolute values among these subgroups, the correlation between DII and NT-proBNP remained consistent.

Furthermore, our study did not identify a significant association between DII and NT-proBNP levels among individuals with HF in the linear regression model (p > 0.05), although the effect size indicated a considerable difference (β = 73.46, 95% CI: -72.47, 219.39). However, analysis of specific DII quartiles revealed significant associations in the second (Q2) and third quartile (Q3), suggesting that moderate to high levels of dietary inflammation may particularly adverse effects in patients with heart failure. This effect became more apparent in Model II, which adjusts for interaction terms and potential confounders, indicating that correct adjustments are necessary to reveal complex interactions and true effects that may be masked by other factors. Furthermore, the limited sample size (n = 374) of the HF group and the use of weighting adjustments may have limited the representativeness and statistical power of the model, advising a cautious interpretation of these results.

Previous studies have highlighted the association between DII scores and CVD morbidity, emphasizing the significant impact of dietary inflammation on health [50]. Our analysis identifies proteins, carbohydrates, dietary fibers, monounsaturated fatty acids, polyunsaturated fatty acids, vitamins A, β-carotene, vitamin B1, niacin, folate, vitamin E, magnesium, iron, selenium, caffeine, alcohol, and n-3 fatty acids as vital dietary factors impacting NT-proBNP levels. For instance, protein and amino acid supplementation may improve muscle mass and functional capacity in patients with HF. However, evidence is still emerging on its effects across different types of HF, such as HF with preserved ejection fraction (HFpEF) [51,52]. Low-carbohydrate diets have demonstrated positive outcomes for patients with diabetic cardiomyopathy, improving symptoms and overall quality of life associated with HF. In a randomized controlled trial, patients with diabetic cardiomyopathy who followed a low-carbohydrate diet exhibited significant weight and systolic blood pressure reductions over 16 weeks [53]. Moreover, studies indicate the beneficial role of polyunsaturated fatty acids, particularly n-3 fatty acids, in cardiovascular health. Their anti-inflammatory and antioxidant properties are credited with lowering the risk of CVD and HF [54,55]. Dietary fiber, particularly from plant

sources, plays a crucial role in cardiac metabolic stability, contributing to a reduced risk of HF. This benefit is assumed to stem from fiber's ability to modulate inflammatory biomarkers [56].

The role of micronutrients such as iron, magnesium, and selenium in HF prognosis is critical. Iron deficiency, prevalent among patients with HF, has been linked to increased cardiovascular hospitalization rates, particularly in those with HF with reduced fraction(HFrEF) [57,58]. Magnesium's role is pivotal in developing HF [59], with deficiencies exacerbating the condition. Selenium's antioxidant properties are vital for tissue protection against oxidative stress [60]. Vitamins A and β-carotene, thiamine (B1), niacin (B3), folate, and vitamin E are crucial for HF management and overall cardiovascular health, though their specific effects warrant further investigation. The association between alcohol intake and HF risk is nuanced, with some studies suggesting potential cardiovascular benefits from moderate alcohol consumption. though evidence remains inconclusive regarding a direct causal relationship [61,62].

This study identified a positive correlation between the DII and NT-proBNP levels. However, the precise biological mechanisms linking these two factors remain elusive. Furthermore, the cross-sectional nature of our study design precluded inferences about causal relationships, necessitating further longitudinal cohort studies to validate our findings. Moreover, reliance on self-reported dietary data collected through 24-hour dietary recalls introduced the potential for measurement inaccuracies and misclassification, possibly not reflecting habitual dietary intake accurately. The absence of data on cardiac function, such as electrocardiogram or echocardiogram results, constrains our understanding of the association between NT-proBNP levels and subclinical cardiac health. Finally, despite the substantial size of our study population, the possibility of residual confounding due to unmeasured variables remains a limitation that could still influence our findings.

## 5 Conclusion

This study identified a positive correlation between the DII and NT-proBNP levels, suggesting a robust link between pro-inflammatory diets and increased heart failure biomarkers, with implications for dietary modifications in cardiovascular risk management.

## Supporting information

**S1 Table. Comparison of each component of DII scores among all participants.** (DOCX)

**S2 Table. Comparison of each actual dietary component of DII among all participants.** (DOCX)

## Acknowledgments

Our sincere appreciation is directed towards the participants of the NHANES database; their role was indispensable to the study's progress.

## Author Contributions

**Conceptualization:** Teng-Chi Ma.

**Data curation:** Teng-Chi Ma.

**Formal analysis:** Teng-Chi Ma.

**Methodology:** Teng-Chi Ma, Jing Zhou.

**Project administration:** Teng-Chi Ma, Jing Zhou.

**Resources:** Teng-Chi Ma.

**Software:** Teng-Chi Ma, Xin-Lu Liu.

**Supervision:** Jing Zhou.

**Validation:** Teng-Chi Ma, Jing Zhou.

**Visualization:** Teng-Chi Ma, Xin-Lu Liu, Jing Zhou.

**Writing – original draft:** Teng-Chi Ma.

**Writing – review & editing:** Teng-Chi Ma, Feng Gao, Chen-Xi Wang, Qiang Liu, Jing Zhou.

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
