## [Decision Letter · Decision Letter 0]

7 Nov 2023

PONE-D-23-33232Association between Dietary Inflammatory Index and NT-proBNP Levels in US adults: A Cross-Sectional AnalysisPLOS ONE

Dear Dr. Chi,

Thank you for submitting your manuscript to PLOS ONE. After careful consideration, we feel that it has merit but does not fully meet PLOS ONE’s publication criteria as it currently stands. Therefore, we invite you to submit a revised version of the manuscript that addresses the points raised during the review process.

ACADEMIC EDITOR: All issues highlighted by reviewers are required in order to support the conclusions.

We look forward to receiving your revised manuscript.

Kind regards,

Vincenzo Lionetti, M.D., PhD

Academic Editor

PLOS ONE

Journal Requirements:

4. Please ensure that you refer to Figure 1 & 3 in your text as, if accepted, production will need this reference to link the reader to the figure.

Reviewers' comments:

Reviewer's Responses to Questions

**Comments to the Author**

1. Is the manuscript technically sound, and do the data support the conclusions?

Reviewer #1: Yes

Reviewer #2: Partly

2. Has the statistical analysis been performed appropriately and rigorously? 

Reviewer #1: Yes

Reviewer #2: Yes

3. Have the authors made all data underlying the findings in their manuscript fully available?

Reviewer #1: Yes

Reviewer #2: Yes

4. Is the manuscript presented in an intelligible fashion and written in standard English?

Reviewer #1: Yes

Reviewer #2: No

5. Review Comments to the Author

Reviewer #1: There are several critical comments for the authors to address:

1. Why was NT-proBNP chosen as the molecule to study? How does it compare to other potential biomarkers? Are there better indicators?

2. Did the authors look at CAD specifically in its analysis?

3. In the Discussion on page 9, the authors have mistaken gender to be synonymous with sex. This is incorrect. You did not monitor gender. You only collected data on sex. Please correct this paragraph on page 9.

4. The authors found no relationship to CVD but continue to discuss it on the first paragraph on page 10. This could be removed as speculation and is not supported statistically by their own data.

5. I agree whole-heartedly with the limitations identified in lines 312 to 320. They are important, especially the reliability of dietary recalls. Simply identifying the limitations does not remove any of the detrimental significance of these limitations. The authors may want to rebut or at least try to address these limitations in some manner to counter their importance.

Reviewer #2: Large-scale identification of possible relationships between pro- or anti-inflammatory properties of diets and cardiovascular (CV) health, and their understanding, can be of great value for CV risk stratification and to guide prevention strategies by focusing on modifiable risk factors. And for these reasons, this work seems promising as it explores on over ten thousand US adults associations between the DII and the NT-proBNP biomarker, accounting for several confounders and stratifying the diet-related CV risks by age groups, gender, ethnicity and chronic diseases.

A number of concerns limits my enthusiasm in endorsing this work, which appears to be a very preliminary version of a publishable article. Major concerns involve many methodological aspects and the logic of analyses; minor concerns involve aspects such as form of writing, contents, organization of contents.

MAJOR REVISIONS

- In “Study Design” section (Line 64): please specify characteristics of the study participants, specifically: minimum and maximum age; number of subjects in the age groups <50 years, 50-75 years, >75 years, divided by gender and ethnicity; inclusion and exclusion criteria based on NT-proBNP and dietary measurements, lifestyle factors, health status and pregnancy status. Reference to Figure 1 is missing in the main text.

- In “Assessment of NT-proBNP” section (Line 83): measurements of NT-proBNP in serum samples were performed after minimum 14 years, this may raise concerns about storage practices and samples integrity. Please specify samples storage method and conditions. Then, indicate the step-by-step protocol for dosing NT-proBNP.

- In “Assessment of Dietary Inflammatory Index” section (Line 91): starting from the 24-hour dietary recall survey, describe in details data gathering, encoding and refinement process. Please, list all the 27 dietary factors used in the DII and describe how it was possible to calculate their intakes (frequencies of consumption, portions, reference nutritional databases). Describe how the DII is structured and the criteria for which the DII score was obtained. Describe the clinical significance of the division into quartiles of DII scores.

- Section that should be added:

- “Assessment of Food Groups and Nutrients Intake” section: it is important to present information on individual food groups, foods and nutrients intake of the study participants, to evaluate dietary intake contributions on relationships between DII and NT-proBNP levels. Please, add related analyses in the “Stratified Analysis” section (Line 201).

- In “Baseline Characteristics of Patients” section (Line 158): p-values in Table 1 refer to inter-quartile differences, therefore the statements from line 164 to 167 require a post-hoc analysis to be statistically supported. From line 162 to 164 ("In patients with...pg/mL.") the sentences refer to Table 2 and not to Table 1. In line 168 the p-value related to C-reactive protein corresponds to <0.001 and not to <0.01 (as shown in Table 2). Tables 1 and 2 present chaotic content and poor captions. Please implement them to ensure a better understanding of the tables.

- In “Relationships of Dietary Inflammatory Index (DII) and NT-proBNP” section (Line 173): in lines 175 and 177 the reported p-value is wrong (see Table 3). Concerning the sentence in lines 191-192, see the comment related to lines 164-167.

- In “Discussion” chapter (Line 217): please, enrich discussion contents by adding a greater number of comparisons between results obtained in your study and those from other studies. Moreover, deepen the discussion regarding negative results obtained. Deepen the influence of dietary factors constituting the DII on cholesterol, triglycerides and glycohemoglobin serum parameters.

MINOR REVISIONS

- In “Abstract” (Line 11): in line 17, the statement "multivariable adjusted analyses" is partly correct. Specify that "non-adjusted analysis" was also performed.

- In “Introduction” chapter (Line 33): from lines 34 to 37 the content is extremely generic and does not provide useful information for understanding the study. From lines 39 to 43 concepts are expressed in a redundant way. Please, add other useful indexes to evaluate associations between the effects of dietary patterns and cardiovascular health. In Lines 50-51, please specify what is meant by "changes in ventricular wall stress". From line 57 to 60 references are missing.

- In “Methods” and “Results” chapters (Line 63/156): choose the term "gender" instead of "sex" in the main text and tables; choose between the terms "unadjusted model" or "crude model" and use the one chosen all the time. Review the table titles.

- In “Discussion” chapter (Line 217): choose the terms "men" and "women" instead of "males" and "females". From line 271 to 273 and from line 278 to 280 references are missing.

- Please go through correction of all typos.

6. PLOS authors have the option to publish the peer review history of their article (what does this mean?). If published, this will include your full peer review and any attached files.

Reviewer #1: No

Reviewer #2: No

---

## [Author Response · Author response to Decision Letter 0]

28 Feb 2024

Dear Editor and Reviewer(s).

We appreciate the opportunity allowing us to revise our manuscript and thanks for reviewers’ constructive comments and suggestions. We would like to submit our revised manuscript, entitled“Association between Dietary Inflammatory Index and NT-proBNP Levels in US adults: A Cross-Sectional Analysis”(Manuscript ID: PONE-D-23-33232) for consideration for publication. In the revised manuscript.we have carefully addressed all comments and questions raised by the reviewer(s) point-by-point. We greatly appreciate your time and efforts to improve our manuscript for publication.

Reviewer1

1.Why was NT-proBNP chosen as the molecule to study? How does it compare to other potential biomarkers? Are there better indicators?

Reply1: Dear Reviewer, thank you for your valuable feedback. With the continuous improvement in laboratory testing capabilities, an increasing number of sensitive and specific cardiac biomarkers are being applied clinically. Laboratory tests for cardiac biomarkers are categorized into: markers of myocardial injury, cardiac function markers, inflammatory markers, and markers of myocardial fibrosis, among others. NT-proBNP is a key biomarker of cardiac function, capable of reflecting cardiac stress and dysfunction, and is widely used in clinical practice as a marker of cardiac function. Recent literature has increasingly confirmed that BNP is a "gold standard" for diagnosing chronic heart failure and assessing cardiac function, particularly in differentiating cardiogenic heart failure from pulmonary diseases[1, 2]. Compared to other biomarkers for heart failure, NT-proBNP has high specificity and sensitivity. A negative BNP result can be used to exclude the diagnosis of left ventricular heart failure. Furthermore, the level of NT-proBNP can assess the severity of heart failure and is an independent risk factor for cardiovascular events within one year in severe heart failure cases[3]. Given that this study explores the relationship between dietary inflammation and cardiovascular health, the association of NT-proBNP with inflammatory states makes it an ideal choice for investigating this relationship.

2.Did the authors look at CAD specifically in its analysis?

Reply2: Thank you for your valuable feedback. In our initial manuscript, our analysis was primarily focused on cardiovascular diseases (CVD). However, your query has highlighted the necessity for a more detailed exploration of heart failure (HF). Therefore, in the revised version, we plan to include an additional weighted multivariate regression analysis specifically addressing HF, to provide deeper insights. Moreover, we will add related unweighted analyses in supplementary materials to further enrich our study and have updated and recreated relevant figures to enhance our presentation.

3.In the Discussion on page 9, the authors have mistaken gender to be synonymous with sex. This is incorrect. You did not monitor gender. You only collected data on sex. Please correct this paragraph on page 9.

Reply3: Thank you for pointing out the inaccurate terminology used in the discussion section on page 9. We acknowledge the incorrect use of "sex" and "gender" within our study, as we indeed only collected data on the gender of participants. Therefore, we will correct this section in the revised manuscript to ensure the accuracy and clarity of our terminology. 

We marked the changes in red in the revised manuscript.These comments are all valuable and enable us to greatly improve thequality ofour manuscript. We tried our best to improve the manuscript.These changes will not influence the content and framework of thepaper. We appreciate for Editors/Reviewers’warm work earnestly. And hope that the corrections will meet with approval. thank you very much for your comments and suggestions.

4.The authors found no relationship to CVD but continue to discuss it on the first paragraph on page 10. This could be removed as speculation and is not supported statistically by their own data.

Reply4: Thank you for your insightful comments, which have inspired new considerations in our approach. Indeed, our data did not demonstrate a significant relationship with cardiovascular disease (CVD). Therefore, we plan to revise the content on page 10 accordingly. We will include additional analyses on heart failure, especially given that our findings were not pronounced in patients with coronary artery disease but were present in those with heart failure (particularly in unweighted cases). We have reanalyzed our data and recreated the relevant tables and figures. We will also discuss how these results vary across different populations to provide a more comprehensive perspective. Thank you for your valuable input, which has significantly enhanced the quality of our manuscript.

5. I agree whole-heartedly with the limitations identified in lines 312 to 320. They are important, especially the reliability of dietary recalls. Simply identifying the limitations does not remove any of the detrimental significance of these limitations. The authors may want to rebut or at least try to address these limitations in some manner to counter their importance.

Reply5: Thank you for highlighting the limitations identified in our study, particularly concerning the reliability of dietary recall. We acknowledge the impact that the limitations regarding dietary recall reliability may have on our findings. While these limitations cannot be completely overcome in the present study, we have endeavored to mitigate their impact through rigorous statistical methods and multivariate analyses. In future research, we plan to employ more precise methods of dietary data collection, such as prospective dietary logs, to enhance the accuracy and reliability of the data.

Reviewer2

1. In “Study Design” section (Line 64): please specify characteristics of the study participants, specifically: minimum and maximum age; number of subjects in the age groups <50 years, 50-75 years, >75 years, divided by gender and ethnicity; inclusion and exclusion criteria based on NT-proBNP and dietary measurements, lifestyle factors, health status and pregnancy status. Reference to Figure 1 is missing in the main text. 

Reply6: Dear Reviewer, Thank you very much for your suggestions and guidance. Regarding the detailed explanation of the characteristics of the study participants in the "Study Design" section as you mentioned, we now provide the following additional information in response to your request:

Characteristics of participants: This study integrated data from three NHANES cycles (1999–2004) to evaluate NT-proBNP levels in participants at a mobile examination center, with blood samples preserved for future analysis. After excluding participants for various reasons, including out-of-range NT-proBNP levels (n=365), incomplete dietary data (n=528), self-reported pregnancy (n=627), and absence of HF diagnosis data (n=43), the study population was narrowed down to 10,766 individuals, comprising 5,475 women and 5,313 men. In the women cohort, 2552 participants were under 50, 2117 were aged between 50 and 75, and 784 were over 75. Among the men, 2426 were younger than 50, 2159 were within the 50 and 75 age range, and 728 were older than 75. Additionally, we noted the absence of a reference to Figure 1 in the main text. We will cite Figure 1 in the appropriate section of the revised manuscript to better aid readers in understanding the study design and participant screening process.

2. In “Assessment of NT-proBNP” section (Line 83): measurements of NT-proBNP in serum samples were performed after minimum 14 years, this may raise concerns about storage practices and samples integrity. Please specify samples storage method and conditions. Then, indicate the step-by-step protocol for dosing NT-proBNP.

Reply7:Thank you for your attention to the assessment of NT-proBNP in our study. Following your suggestion, we have revised this section's description and uploaded the relevant original experimental documents as supplementary files[4]. We have revised the description to read: Serum samples from the 1999-2004 NHANES cycles were analyzed for NT-proBNP levels, with the assays conducted between 2018 and 2020 at the University of Maryland School of Medicine’s laboratory, under the leadership of Dr. Roche Diagnostics. The NHANES Biospecimen Program provided protocols for sample handling, detailing the processes for processing and storage. NT-proBNP measurements were performed using the Roche Cobas e601 analyzer (Elecsys, Roche Diagnostics), adhering to established detection limits (5 pg/mL to 35,000 pg/mL) and exhibiting a coefficient of variation of 3.1% at a low concentration (46 pg/mL) and 2.7% at a high concentration (32,805 pg/mL). A consistent detection threshold was employed to ensure the accuracy and reliability of the measurements. Additionally, prior research has demonstrated the stability of NT-proBNP in samples stored for extended periods[5-7].Thank you for your suggestion, which has now made this section of the manuscript clearer.

3. In “Assessment of Dietary Inflammatory Index” section (Line 91): starting from the 24-hour dietary recall survey, describe in details data gathering, encoding and refinement process. Please, list all the 27 dietary factors used in the DII and describe how it was possible to calculate their intakes (frequencies of consumption, portions, reference nutritional databases). Describe how the DII is structured and the criteria for which the DII score was obtained. Describe the clinical significance of the division into quartiles of DII scores, and“Assessment of Food Groups and Nutrients Intake” section: it is important to present information on individual food groups, foods and nutrients intake of the study participants, to evaluate dietary intake contributions on relationships between DII and NT-proBNP levels. Please, add related analyses in the “Stratified Analysis” section (Line 201).

Reply8: Thank you for your attention in our study. In this study, participants' dietary intake data were collected via a 24-hour dietary recall method. Through the Mobile Examination Center (MEC), detailed records of the food and beverages consumed by participants in the 24 hours preceding the interview were documented. The reported dietary components were subsequently quantified using the USDA Food and Nutrient Database for Dietary Studies to determine the content of specific nutrients.

DII Components

this research was based on 27 dietary components known to modulate inflammation, including energy, protein, carbohydrates, fiber, total fats, saturated fats, monounsaturated and polyunsaturated fatty acids, cholesterol, a range of vitamins and minerals including vitamin A, β-carotene, thiamin, riboflavin, niacin, vitamin B6, folic acid, vitamin B12, vitamin C, vitamin E, magnesium, iron, zinc, and selenium, caffeine, alcohol, and n-3 and n-6 fatty acids

DII Calculation Method

Each participant's dietary intake was matched to a comprehensive database encompassing global average daily intake figures and their standard deviations for each dietary component across 45 populations worldwide. The intake of each nutrient was contrasted with its global average and subsequently divided by the standard deviation to calculate the Z-scores for each nutrient. These Z-scores were converted into percentiles, centered around zero by doubling and then subtracting one, yielding scores from -1 to +1. Each centered score was multiplied by its respective dietary component’s inflammation effect score to calculate specific DII scores for each food parameter. The overall DII score for an individual was calculated by aggregating these specific DII scores.

Clinical Significance

The division into quartiles of DII scores was utilized to reflect levels of dietary inflammation. Higher DII scores typically indicate a pro-inflammatory diet, while lower DII scores suggest an anti-inflammatory diet. Previous research, as well as our study, has demonstrated significant associations between the DII and inflammatory markers such as Klotho and C-reactive protein (CRP), confirming the clinical relevance of DII scoring[8].

Additionally, following the review of your comments, we recognized the importance of providing more information on food categories and nutrient intakes. We plan to include a lasso regression of the original data constituting the DII in the revised manuscript, and Figure 4 will display the association of each component with NT-proBNP. We will also further elucidate in the text why we believe that focusing on the overall DII score is appropriate. While decomposing the DII components for individual analysis may have its value in certain contexts, in the backdrop of our study, we believe that concentrating on the overall DII score better captures the holistic impact of dietary patterns on inflammation and heart health. This aligns with the original intent of our study design, which is to assess the impact of overall dietary patterns rather than single foods or nutrients on heart health.

We believe such additions will offer a more comprehensive view to our readers, aiding in their understanding. In summary, we look forward to addressing your comments more comprehensively through these revisions, thereby enhancing the depth and breadth of our study.

4.In “Baseline Characteristics of Patients” section (Line 158): p-values in Table 1 refer to inter-quartile differences, therefore the statements from line 164 to 167 require a post-hoc analysis to be statistically supported. From line 162 to 164 ("In patients with...pg/mL.") the sentences refer to Table 2 and not to Table 1. In line 168 the p-value related to C-reactive protein corresponds to <0.001 and not to <0.01 (as shown in Table 2). Tables 1 and 2 present chaotic content and poor captions. Please implement them to ensure a better understanding of the tables.

Reply9: Thank you for your thorough review and valuable suggestions. Regarding the issues raised with Tables 1 and 2, we have made the following amendments and restructuring to the description of the results section:

Baseline Characteristics of Participants:

Table 1 presents the baseline characteristics of the study population, divided into quartiles (Q1-Q4) based on their Dietary Inflammatory Index (DII) scores to examine variable distribution. Out of 10,766 participants, 374 reported a history of heart failure (HF). Those with HF had a mean age of approximately 64.77 ± 1.16 years, with a gender distribution of 48.26% women and 51.74% men. Racial and ethnic backgrounds predominantly included non-Hispanic whites (76.87%), with the remainder being Mexican Americans (3.23%), non-Hispanic blacks (10.76%), and other races (9.14%). Chronic diseases were prevalent in the HF group, with chronic kidney disease (CKD) present in 51.01%, hypertension in 70.48%, hyperlipidemia in 86.71%, and diabetes in 34.18%.

For the 9,392 participants without a history of HF, stratification by DII scores revealed distinct quartiles, showing an increasing percentage of women and a decreasing percentage of men with higher DII quartiles. The racial and ethnic distribution among the non-HF participants closely mirrored that of the HF cohort. The prevalence of chronic diseases such as CKD, hypertension, and hyperlipidemia increased with higher DII quartiles. Weighted linear regression analyses identified significant associations between DII and demographic variables, including age, gender, and race or ethnicity (gender p < 0.001, race or ethnicity p < 0.001), showing distinct variations across DII quartiles. A significant increase in body mass index (BMI) was observed with higher DII quartiles (p < 0.001), though waist circumference remained relatively constant across the groups (p = 0.370).

Table 2 highlights that, in patients with HF, the mean level of N-terminal pro b-type natriuretic peptide (NT-proBNP) was significantly elevated, recorded at 1035.53 ± 113.63 pg/mL. Other cardiovascular risk factors measured included albumin, high-density lipoprotein cholesterol (HDL-C), low-density lipoprotein cholesterol (LDL-C), triglycerides, and total cholesterol, with mean values of 4.16 ± 0.02 g/dL, 1.24 ± 0.03 mg/dL, 2.93 ± 0.08 mg/dL, 171.50 ± 5.62 mg/dL, 

---

## [Decision Letter · Decision Letter 1]

19 Mar 2024

PONE-D-23-33232R1Association between Dietary Inflammatory Index and NT-proBNP Levels in US adults: A Cross-Sectional AnalysisPLOS ONE

Dear Dr. Chi,

Thank you for submitting your manuscript to PLOS ONE. After careful consideration, we feel that it has merit but does not fully meet PLOS ONE’s publication criteria as it currently stands. Therefore, we invite you to submit a revised version of the manuscript that addresses the points raised during the review process.

**ACADEMIC EDITOR: **All issues raised by expert reviewer are required.

We look forward to receiving your revised manuscript.

Kind regards,

Vincenzo Lionetti, M.D., PhD

Academic Editor

PLOS ONE

Journal Requirements:

Reviewers' comments:

Reviewer's Responses to Questions

**Comments to the Author**

1. If the authors have adequately addressed your comments raised in a previous round of review and you feel that this manuscript is now acceptable for publication, you may indicate that here to bypass the “Comments to the Author” section, enter your conflict of interest statement in the “Confidential to Editor” section, and submit your "Accept" recommendation.

Reviewer #2: All comments have been addressed

2. Is the manuscript technically sound, and do the data support the conclusions?

Reviewer #2: Yes

3. Has the statistical analysis been performed appropriately and rigorously? 

Reviewer #2: No

4. Have the authors made all data underlying the findings in their manuscript fully available?

Reviewer #2: Yes

5. Is the manuscript presented in an intelligible fashion and written in standard English?

Reviewer #2: No

6. Review Comments to the Author

Reviewer #2: It is very appreciable to observe that the authors have done a substantial job, where the new version of the manuscript is much improved compared to the previous one and adequate to the comments raised by the reviewers. However, there are still a number of major and minor critical aspects that authors must address, to make this work more rigorous, transparent, easier to understand and less chaotic.

MAJOR REVISIONS:

- Please explain how you were able to detect in non-HF patients (Table 1) levels of NT-proBNP from 95.06±7 pg/dL (so 0,95 pg/mL) to 129.91±7.04 pg/dL (so 1.2 pg/mL) if you report that minimum detection level is set at 5 pg/mL. Please, check unit of measure for NT-proBNP in HF subjects. Please also be aware that the unit of measure you report for HDL and LDL levels in Table 1 is wrong; not being aware of their meaning would be a serious concern for publication.

- Student's t test and chi-square test for comparing less than 3 groups at a time. I therefore believe that for the analyzes reported in Table 1 and Table 2 for non-HF subjects, one-way ANOVA (normally distributed data) or Kruskal-Wallis (non-normally distributed data) would be appropriate. Then, please use these suggested tests in your analyses.

- Please, in partial correlation analyses use Spearman’s coefficient for non-normally distributed data or Pearson’s coefficient for normally distributed data.

- Please, in your adjusted analyses use only those variables underlying significant interactions (e.g, age, gender, CKD, hypertension, diabetes).

MINOR REVISIONS:

- Abstract: “Stratified analyses were then conducted to examine the associations within specific subgroups to identify differential effects of the DII on NT-proBNP levels across diverse populations.”; “In individuals without heart failure, a unit increase in the DII was significantly associated with an increase in NT-proBNP levels.”; “Significant interactions were observed in subgroup analyses by age, gender, hypertension, and diabetes.” Please, for a quick understanding by riders, explicitly indicate what these specific subgroups and diverse populations, and a unit increase in the DII, and subgroups analyses are.

- It is reported in the abstract and in line 62 that your analyses refer to data from the NHANES and collected in 1999-2002; however, in lines 77 and 88 you indicate data from three NHANES cycles 1999-2004; so, please indicate the start/end date (year) of each cycle.

- Please add data indicating the quantification (mean and SD) of each dietary component of the DII, and DII scores (continuous data) and DII quartiles, for HF and non-HF subjects.

- Please indicate maximum and minimum values of the DII score; and indicate the inflammation effect for each dietary component.

- Please change the section title “Assessment of covariates” in “Assessment of study participants’ characteristics”; please delete the first sentence in lines 119-120.

- Please, explain the reasons of using DII quartiles instead of DII continuous data in your analyses. Please, indicate that by using DII continuous data it did not produce significant results, or add these results. Please, indicate that using DII quartiles reduced inter-individual variability leading to significant findings.

- In line 143-144-145: please, don’t use the term segmented. Also, the cut off for DII quartiles you report are wrong (Q1: DII ≤ 144 0.753; Q2: 0.753 < DII ≤ 2.172; Q3: 2.172 < DII ≤ 3.193; Q4: DII 3.193). Please don’t use the term variances.

- In line 156: BMI categories cut-off are usually different from those you report, please specify the reason why of these cut-offs you chose.

- Please add in Table 1 result from comparison between DII scores and NT-proBNP levels (continuous data) also for non-HF subjects.

-In lines 180-181: “stratification by DII scores revealed distinct quartiles.” Please, be explain the term distinct.

- In line 189: “NT-proBNP level was significantly elevated”; please indicate that are elevated in comparison to NT-proBNP in non-HF subjects, and provide references for this statement.

-In lines 210-212: your statement sounds speculative, only results from model I indicate significance.

- Please indicate in all tables the specific statistical test used, and results as mean and SD were appropriate.

- In line 265: “incorporating 27 dietary factors and three covariates”; please, indicate why you use those covariates, maybe you would meant you used covariates of model I.

- In line 272: please change your sentence in “Out of n=27 dietary variables constituting the DII scores, n=17 show an interaction to NT-proBNP levels in whole population, such as: protein, etc.”.

- In line 290: use “in preserving cardiovascular health” instead of management; then, specify from where is obtained the value of 14.32 pg/dL.

- In line 49; 90-91; 111; 116 reference is missing.

- In line 52: “The DII objective” is correct and “The DII’s objective” is not; in line 142: “the study complex” is correct and “the study’s complex” is not; in line 188: “Table 2 highlights” is correct and “Table 2 highlighted” is not.

- In lines 77-78: “After excluding participants for various reasons”; please, be specific and don’t use any form such as “various reasons”.

- In line 90: who is Dr. Roche Diagnostics?

- Please go throughout all typos, and revise all tables and figures in an intelligible fashion.

7. PLOS authors have the option to publish the peer review history of their article (what does this mean?). If published, this will include your full peer review and any attached files.

Reviewer #2: No

---

## [Author Response · Author response to Decision Letter 1]

16 Apr 2024

Dear Editor and Reviewer.

We are grateful for the opportunity to revise our manuscript entitled "Association between Dietary Inflammatory Index and NT-proBNP Levels in US adults: A Cross-Sectional Analysis" (Manuscript ID: PONE-D-23-33232) and would like to resubmit it for consideration. We have carefully addressed all comments and questions raised by the reviewer(s) in a point-by-point manner. Thank you for your time and efforts in helping us improve our manuscript for publication.

1.Please explain how you were able to detect in non-HF patients (Table 1) levels of NT-proBNP 

from 95.06±7 pg/dL (so 0,95 pg/mL) to 129.91±7.04 pg/dL (so 1.2 pg/mL) if you report that minimum detection level is set at 5 pg/mL. Please, check unit of measure for NT-proBNP in HF subjects. Please also be aware that the unit of measure you report for HDL and LDL levels in Table 1 is wrong; not being aware of their meaning would be a serious concern for publication.

We sincerely appreciate your thorough review and the critical issues you have brought to our attention. Your comments have been invaluable in identifying areas that require clarification and correction, ultimately enhancing the quality of our manuscript.

Upon careful re-examination of the data, we acknowledge the discrepancies in the NT-proBNP levels reported in Table 1. We will rectify this error by ensuring all values are consistently expressed in pg/mL, in line with the assay's minimal detection level of 5 pg/mL. Furthermore, we will meticulously review the entire manuscript to guarantee that all NT-proBNP data is accurately presented and discussed.

Regarding the incorrect units for HDL and LDL levels, we deeply regret this oversight and understand the potential implications it may have on the interpretation of our findings. We will promptly amend the table to reflect the correct units and will diligently verify that all biomarker data throughout the manuscript is accurately represented.

We are committed to maintaining the highest standards of scientific integrity and accuracy in our research. Your feedback has been a crucial reminder of the importance of rigorous data presentation and interpretation. Rest assured that we will implement more stringent review processes to prevent such errors from occurring in our future work.

2.Student's t test and chi-square test for comparing less than 3 groups at a time. I therefore believe that for the analyzes reported in Table 1 and Table 2 for non-HF subjects, one-way ANOVA (normally distributed data) or Kruskal-Wallis (non-normally distributed data) would be appropriate. Then, please use these suggested tests in your analyses.Please indicate in all tables the specific statistical test used, and results as mean and SD were appropriate.

Thank you for your meticulous review and valuable suggestions. In response to your comments regarding the appropriateness of statistical methods, we have reconsidered our analysis strategy. Initially, Table 1 did indeed involve four DII groups, for which one-way ANOVA or the Kruskal-Wallis test is generally recommended to assess intergroup differences. However, following your guidance and considering the specific distribution of our data, we have now revised our grouping method, simplifying it into two main categories: those with a history of heart failure and those without (Non-HF). With this new grouping strategy, for the comparisons between these two groups, we employed Student's t-test (for normally distributed continuous data) and the chi-square test (for categorical data). These methods are more suitable for comparing mean differences and proportions between two independent samples. All statistical results have been updated to ensure the correct application of statistical tests, and each test's p-value has been clearly indicated in the tables. The results are now reported as mean ± standard deviation for normally distributed data and as actual numbers and weighted percentages for categorical variables.

We appreciate your guidance in refining our statistical approach and data presentation. Your suggestions have undoubtedly strengthened the manuscript, and we are grateful for your thorough review.

3.- Please, in your adjusted analyses use only those variables underlying significant interactions (e.g, age, gender, CKD, hypertension, diabetes).

Thank you very much for your detailed review and suggestions regarding our analysis method. We agree with your point that only those variables that show significant interaction (such as age, gender, CKD, hypertension, and diabetes) should be used in the adjustment of the analysis.

Based on your suggestion, we plan to redo our analysis. In the revised Model II, we will limit the adjusting variables to age, gender, ethnicity, CKD, hypertension, and diabetes. This step will help us understand more accurately the relationship between these variables and the study outcomes and strengthen our conclusions.

The relevant tables will be updated to reflect these adjustments, and more precise data interpretations will be provided in the paper. We believe these changes will enhance the quality and credibility of our study.

4.Please, in partial correlation analyses use Spearman’s coefficient for non-normally distributed data or Pearson’s coefficient for normally distributed data.

Thank you for your thoughtful suggestion regarding the use of Spearman's or Pearson's correlation coefficients in our analyses, depending on the normality of the data distribution. We appreciate your attention to this important methodological consideration.

However, we would like to respectfully point out that the complex nature of the NHANES dataset, which involves intricate sampling designs and weighting procedures, poses limitations to the straightforward application of simple correlation analyses. These methods often do not account for the sample weights, which are crucial in ensuring the representativeness of the findings. Previous NHANES literature rarely uses Spearman's coefficient for non-normally distributed data or Pearson's coefficient for normally distributed data[1-4].

Moreover, in our study, we have prioritized the use of multivariate regression models to control for multiple confounding factors. Given the presence of numerous potential influencing variables, relying solely on correlation analyses (such as Pearson's or Spearman's) may not suffice to uncover the true relationships between the variables of interest. Multivariate analyses, on the other hand, provide a more comprehensive perspective by simultaneously considering the effects of multiple factors.

As demonstrated in "Table 2. Relationship between Dietary Inflammatory Index and NT-proBNP," we have conducted thorough multivariate modeling to examine the associations between the DII and NT-proBNP levels, while adjusting for relevant covariates such as age, gender, race/ethnicity, hypertension, diabetes, and chronic kidney disease. This approach allows for a more robust assessment of the relationships, taking into account the complex interplay of various factors.

We hope that this explanation clarifies our rationale for the analytical methods employed in our study. Rest assured that we have carefully considered the appropriateness of the statistical techniques used, given the specific characteristics of the NHANES dataset and the research questions at hand.

Thank you once again for your valuable input and for engaging in this scientific discourse.

5.Abstract: “Stratified analyses were then conducted to examine the associations within specific subgroups to identify differential effects of the DII on NT-proBNP levels across diverse populations.”; “In individuals without heart failure, a unit increase in the DII was significantly associated with an increase in NT-proBNP levels.”; “Significant interactions were observed in subgroup analyses by age, gender, hypertension, and diabetes.” Please, for a quick understanding by riders, explicitly indicate what these specific subgroups and diverse populations, and a unit increase in the DII, and subgroups analyses are.

Thank you for your insightful comments and suggestions regarding our manuscript. We appreciate the opportunity to clarify the details of our analyses and findings for a more comprehensive understanding. We have revised our abstract to explicitly detail the specific subgroups and diverse populations investigated, as well as to clarify the implications of a unit increase in the DII and our approach to subgroup analyses. The amendments are as follows:

Abstract

Background: With cardiovascular diseases standing as a leading cause of mortality worldwide, the interplay between diet-induced inflammation, as quantified by the Dietary Inflammatory Index (DII), and heart failure biomarker NT-proBNP has not been investigated in the general population.

Methods: This study analyzed data from the National Health and Nutrition Examination Survey (NHANES) 1999-2004, encompassing 10,766 individuals. The relationship between the DII and NT-proBNP levels was evaluated through multivariable-adjusted regression models. To pinpoint crucial dietary components influencing NT-proBNP levels, the LASSO regression model was utilized. Stratified analyses were then conducted to examine the associations within specific subgroups to identify differential effects of the DII on NT-proBNP levels across diverse populations.

Results: In individuals without heart failure, a unit increase in the DII was significantly associated with an increase in NT-proBNP levels. Specifically, NT-proBNP levels rose by 9.69 pg/mL (95% CI: 6.47, 12.91; p < 0.001) without adjustments, 8.57 pg/mL (95% CI: 4.97, 12.17; p < 0.001) after adjusting for demographic factors, and 5.54 pg/mL (95% CI: 1.75, 9.32; p = 0.001) with further adjustments for health variables. In participants with a history of heart failure, those in the second and third DII quartile showed a trend towards higher NT-proBNP levels compared to those in the lowest quartile, with increases of 717.06 pg/mL (95% CI: 76.49-1357.63, p = 0.030) and 855.49 pg/mL (95% CI: 156.57-1554.41, p = 0.018). Significant interactions were observed in subgroup analyses by age (<50: β=3.63, p=0.141; 50-75: β=18.4, p<0.001; >75: β=56.09, p<0.001), gender (men: β=17.82, p<0.001; women: β=7.43, p=0.061),hypertension (β=25.73, p<0.001) and diabetes (β=38.94, p<0.001).

Conclusion: This study identified a positive correlation between the DII and NT-proBNP levels, suggesting a robust link between pro-inflammatory diets and increased heart failure biomarkers, with implications for dietary modifications in cardiovascular risk management.

Thank you for your careful review of our manuscript and for bringing this discrepancy to our attention. We sincerely apologize for the oversight in our reporting regarding the NHANES data cycles.

6. It is reported in the abstract and in line 62 that your analyses refer to data from the NHANES and collected in 1999-2002; however, in lines 77 and 88 you indicate data from three NHANES cycles 1999-2004; so, please indicate the start/end date (year) of each cycle.

Thank you for your careful review of our manuscript and for bringing this discrepancy to our attention. We sincerely apologize for the oversight in our reporting regarding the NHANES data cycles.

We have rectified this error in the revised manuscript. To clarify, the NHANES data utilized in our analyses were collected across three cycles: 1999-2000, 2001-2002, and 2003-2004. We have updated the relevant sections of the manuscript to accurately reflect the start and end dates of each cycle.

We appreciate your diligence in ensuring the accuracy and integrity of our research findings. Please let us know if you require any further clarification or if there are any additional concerns.

7. Please add data indicating the quantification (mean and SD) of each dietary component of the DII, and DII scores (continuous data) and DII quartiles, for HF and non-HF subjects.Please add data indicating the quantification (mean and SD) of each dietary component of the DII, and DII scores (continuous data) and DII quartiles, for HF and non-HF subjects.Please indicate maximum and minimum values of the DII score; and indicate the inflammation effect for each dietary component.

Thank you for your detailed review and for the suggestion to provide a more detailed presentation of the data. In response to your request, we have added comprehensive data detailing the quantification of each dietary component within the Dietary Inflammatory Index as well as the DII scores (continuous data) and DII quartiles, for both HF and non-HF subjects.

During our analysis, we found that the DII scores ranged from a minimum of -4.28 to a maximum of 5.18. Furthermore, the inflammatory effect of each dietary component has also been presented in these tables to offer an intuitive understanding of the impact of each component on the overall DII score. We have meticulously presented this data in tables included in the appendix:

Table S1: Provides the relationship between various dietary components and heart failure status (HF and non-HF), including means and standard deviations.

Table S2: Provides the corresponding data for the DII scores of the actual dietary components related to DII scores with heart failure status.

We believe these additional data and tables fully address your query and enhance the transparency and credibility of our research. 

8.In line 290: use “in preserving cardiovascular health” instead of management; then, specify from where is obtained the value of 14.32 pg/dL.

Thank you for your valuable suggestions. I have made the change on line 290, replacing "management" with "in preserving cardiovascular health" as you directed. Regarding the source of the value of 14.32 pg/dL, this figure is derived from the pooled population analysis presented in the forest plot of our manuscript. It based on a consolidated analysis of the data from all study participants included. This approach ensures a comprehensive and representative result, providing an accurate estimate that reflects the overall trend.

9.In lines 77-78: “After excluding participants for various reasons”; please, be specific and don’t use any form such as “various reasons”.

Thank you for pointing out the need for specificity in describing the exclusion criteria for our study participants. In accordance with your suggestion, I have revised lines 77-78 of the manuscript to precisely state the reasons for participant exclusion. The revised text is as follows:

"Participants were excluded based on the following criteria: out-of-range NT-proBNP levels (n=365), incomplete dietary data (n=528), self-reported pregnancy (n=627), and absence of heart failure diagnosis data (n=43). Following these exclusions, the study population was narrowed down to 10,766 individuals."With gratitude for your thorough review and guidance.

10.Please change the section title “Assessment of covariates” in “Assessment of study participants’ characteristics”; please delete the first sentence in lines 119-120.

Thank you for your guidance on refining the section titles and content within our manuscript. As per your suggestions, the section title previously named “Assessment of covariates” has been changed to “Assessment of study participants’ characteristics” to more accurately reflect the content of the section.

Additionally, the first sentence in lines 119-120 has been removed in accordance with your instructions. We believe these changes improve the clarity and focus of the manuscript, and we are grateful for your constructive feedback.

11.Please, explain the reasons of using DII quartiles instead of DII continuous data in your analyses. Please, indicate that by using DII continuous data it did not produce significant results, or add these results. Please, indicate that using DII quartiles reduced inter-individual variability leading to significant findings.

Thank you for your interest in the methodological choices of our study. In this research, we opted to use quartiles of the Dietary Inflammatory Index (DII) rather than solely relying on continuous data for analysis. This approach is principally because quartiles can more effectively reveal the relationships between variables, especially w

---

## [Decision Letter · Decision Letter 2]

9 May 2024

Association between Dietary Inflammatory Index and NT-proBNP Levels in US adults: A Cross-Sectional Analysis

PONE-D-23-33232R2

Dear Dr. Zhou,

We’re pleased to inform you that your manuscript has been judged scientifically suitable for publication and will be formally accepted for publication once it meets all outstanding technical requirements.

Kind regards,

Vincenzo Lionetti, M.D., PhD

Academic Editor

PLOS ONE

Additional Editor Comments (optional):

Reviewers' comments:

Reviewer's Responses to Questions

**Comments to the Author**

1. If the authors have adequately addressed your comments raised in a previous round of review and you feel that this manuscript is now acceptable for publication, you may indicate that here to bypass the “Comments to the Author” section, enter your conflict of interest statement in the “Confidential to Editor” section, and submit your "Accept" recommendation.

Reviewer #2: All comments have been addressed

2. Is the manuscript technically sound, and do the data support the conclusions?

Reviewer #2: Yes

3. Has the statistical analysis been performed appropriately and rigorously? 

Reviewer #2: Yes

4. Have the authors made all data underlying the findings in their manuscript fully available?

Reviewer #2: Yes

5. Is the manuscript presented in an intelligible fashion and written in standard English?

Reviewer #2: Yes

6. Review Comments to the Author

Reviewer #2: After the last revision you made, I belive your manuscript could suit for publication. Just, add on table and figure captions the type of statistical analyses used.

7. PLOS authors have the option to publish the peer review history of their article (what does this mean?). If published, this will include your full peer review and any attached files.

Reviewer #2: No

---

## [Editor Report · Acceptance letter]

14 May 2024

PONE-D-23-33232R2 

PLOS ONE

Dear Dr. Zhou, 

I'm pleased to inform you that your manuscript has been deemed suitable for publication in PLOS ONE. Congratulations! Your manuscript is now being handed over to our production team.

Kind regards, 

on behalf of

Prof. Vincenzo Lionetti 

Academic Editor

PLOS ONE